# The Presence of Wodginite in Lithium–Fluorine Granites as an Indicator of Tantalum and Tin Mineralization: A Study of Abu Dabbab and Nuweibi Massifs (Egypt)

Viktor I. Alekseev and Ivan V. Alekseev *

Faculty of Geological Prospecting, Saint-Petersburg Mining University, 2, 21st Line, St. Petersburg 199106, Russia; alekseev_vi@pers.spmi.ru
* Correspondence: alekseev_iv@pers.spmi.ru

**Abstract:** This study examines the accessory wodginite and the discovery of titanium-bearing wodginite and Fe and Ti-bearing wodginite in lithium-fluorine granites from the Abu Dabbab and Nuweibi massifs in Eastern Egypt. The wodginite group's mineral association includes tantalum-bearing cassiterite and tin-bearing tantalum–niobate minerals: tantalite-(Mn), columbite-(Mn), and microlite. Three forms of wodginite crystallization were identified: (1) rims around columbite-(Mn) and tantalite-(Mn) varying from 1.5 to 21.9 μm in thickness, (2) micro-inclusions in cassiterite ranging from 5.4 to 27.0 μm in size, and (3) autonomous crystals measuring 3–124 μm in length. Wodginite in the Nuweibi massif is mainly found in porphyritic granite of late-stage porphyry intrusion. It has a similar composition to the worldwide wodginite of rare-metal granites, but exhibits a lower content of $TiO_2$ (average 0.54%) and is a mineral indicator of rich tantalum ore deposits. In contrast, wodginite in the Abu Dabbab massif is replaced by titanium-bearing wodginite (Ti/(Sn + $^B$Ta + Ti + $Fe^{3+}$) = 0.23) and is associated with Fe and Ti-bearing wodginite. Wodginite and Ti-bearing wodginite are maximally enriched in manganese (Mn/(Mn + $Fe^{2+}$ +Ca) = 0.95), expressed in all intrusive phases of the massif, and are mineral indicators of tantalum-bearing granites with associated cassiterite-quartz mineralization.

**Keywords:** wodginite; titanowodginite; ferrotitanowodginite; lithium-fluorine granite; deposit; tantalum; tin; isomorphism; eastern desert; Egypt

## 1. Introduction

The rapid development of high-technology metallurgy, electronics, and other industrial sectors have determined the growth of a global demand for tantalum. Its main reserves are concentrated in rare-metal pegmatites in Australia, Canada, Ethiopia, Mozambique, and in alluvial deposits and weathering crusts of rare-metal ores in Brazil, Central Africa, and Canada. The economic importance of tin–tantalum deposits in peraluminous lithium-fluorine granites (LFG), such as in Abu Dabbab and Nuweibi in Egypt [1,2], Yichun and Songshugang in China [3], Orlovskoe and Voznesenskoe in Russia, and others, is increasing [4,5]. The study of tantalum-rich rare-metal granites is relevant because it allows us to diversify the sources of tantalum in conditions of growing demand, market instability, and unequal geographic distribution of global raw-material reserves [6,7].

In addition to tantalite, microlite, and tantalum-bearing cassiterite, the main tantalum minerals in pegmatites also include wodginite, discovered in 1963 at the Bernic Lake deposit in Canada, and occurring as an industrial mineral in tantalite–microlite ore [8]. The wodginite group includes wodginite $MnSnTa_2O_8$, titanowodginite $MnTiTa_2O_8$, ferrowodginite $FeSnTa_2O_8$, ferrotitanowodginite $FeTiTa_2O_8$, lithiowodginite $LiTaTa_2O_8$, tantalowodginite $(Mn_{0.5}\square_{0.5})TaTa_2O_8$, and "wolframowodginite" (unconfirmed mineral species) $MnTi(Ta,W)_2O_8$.

Since 2002, wodginite group minerals have been increasingly found in LFGs in China (Yichun, Dajishan) [9–11], Egypt (Abu Dabbab, Nuweibi, Mueilha) [1,12], Spain (Penouta),

Czech Republic (Hub), and Russia (Voznesenskoe, Kester). In these granites, these minerals have small sizes (1–100 μm), and they do not include lithiowodginite or tantalowodginite. Titanowodginite occurs three times more frequently in LFGs than in pegmatites [9,13,14].

The objective and scientific novelty of this article is to study accessory wodginite and the discovery of titanium-bearing wodginite in Li-F granites in Abu Dabbab and Nuweibi massifs in eastern Egypt [15] as indicator minerals of tantalum-bearing granites and accompanying tantalum and tin mineralization.

## 2. Materials and Methods

This article reports the findings of a study conducted by the Department of Mineralogy, Crystallography, and Petrography at St. Petersburg Mining University (SPMU). A total of 55 samples of granites from the Abu Dabbab and Nuweibi tin–tantalum deposits were examined and analyzed at the Center for Collective Usage (CCU) of SPMU using the following methods:

- Leica DM 2500 M (Leica Microsystems, Wetzlar, Germany) microscope for petrographic analysis;
- XRF-1800 (Shimadzu Corporation, Kyoto, Japan) X-ray fluorescence spectrometry for major and trace elements;
- ICPE-9000 (Shimadzu Corporation, Kyoto, Japan) inductively coupled plasma mass spectrometry for rare earth elements (REE) and yttrium (Y).

The analysis allowed for the identification of the least-altered LFGs with typical structures and high Li, Rb, Ta, Nb, Sn, REE, and Y contents; we used 11 samples from the Abu Dabab array and 9 samples from the Nuweibi array.

In transparent, polished, carbon-coated, thin sections of these rocks, an optical search for rare-earth mineral carriers and their analysis was conducted at the Institute of Precambrian geology and geochronology, Russian Academy of Sciences (IPGG RAS), St. Petersburg. The analytical conditions were:

- JEOL JSM-6510LA (JEOL Ltd., Tokyo, Japan) scanning electron microscope with the JED-2200 energy-dispersive spectrometer;
- Accelerating voltage 20 kV, current 1.5 nA;
- ZAF correction method for matrix effects.

Upon detailed examination of the accessory minerals in the LFGs from both arrays, columbite-(Mn) and tantalite-(Mn) with Sn impurities, microlite with U, Pb, Bi, cassiterite with Ta and Nb impurities, zircon with Hf, U, and others were found. Wodginite was identified in both sets of granites as described in this article. The quantitative chemical analysis of wodginite (16 samples, 130 analyses) was evaluated using the JEOL JXA-8230 microprobe with wave spectrometers at the CCU of SPMU. The system worked with an accelerating voltage of 20 kV, a beam current of 100 nA, a spot size of 3 μm, and a counting time of 30 s for peaks and 10 s for background. Matrix effects were corrected using the ZAF method. Standards used were $LiTaO_3$ (Ta Lα), $LiNbO_3$ (Nb Lα), spessartine (Mn Kα), $SnO_2$ (Sn Lα), fayalite (Fe Kα), TiO (Ti Kα), diopside (Ca Kα), Sc metal (Sc Kα), and zircon (Si Kα, Zr Lα, Hf Lα).

Wodginite diagnostics were performed based on the work in [16–18] and the latest review of the mineralogy of the wodginite group [14]. The data were statistically processed taking into account the content distribution of the components [19]. Special attention was paid to the interrelationship of wodginite with other minerals, the anatomy of individuals [20], and the composition and distribution of impurity elements and mineral inclusions in crystals [21]. As practice has shown, ontogenetic studies of accessory minerals can provide important information about the origin and formation conditions of igneous rocks, their ore-bearing potential [22–24].

## 3. Geological Setting

### 3.1. Regional Geology

The study area is located in the central part of the Eastern Desert of Egypt, where a low-lying relief is developed: eroded granite formations that are separated by wadis—dry valley channels with proluvial cassiterite and columbite–tantalite placers. In the early 1970s, as part of the implementation of the Egyptian–Soviet geological exploration program, primary sources of these placers; Sn-Nb-Ta deposits in albite granites of the Igla, Nuweibi, Abu Dabbab, and Hamr Waggat massifs were discovered in the vicinity of the Red Sea coast [25] (Figure 1a).

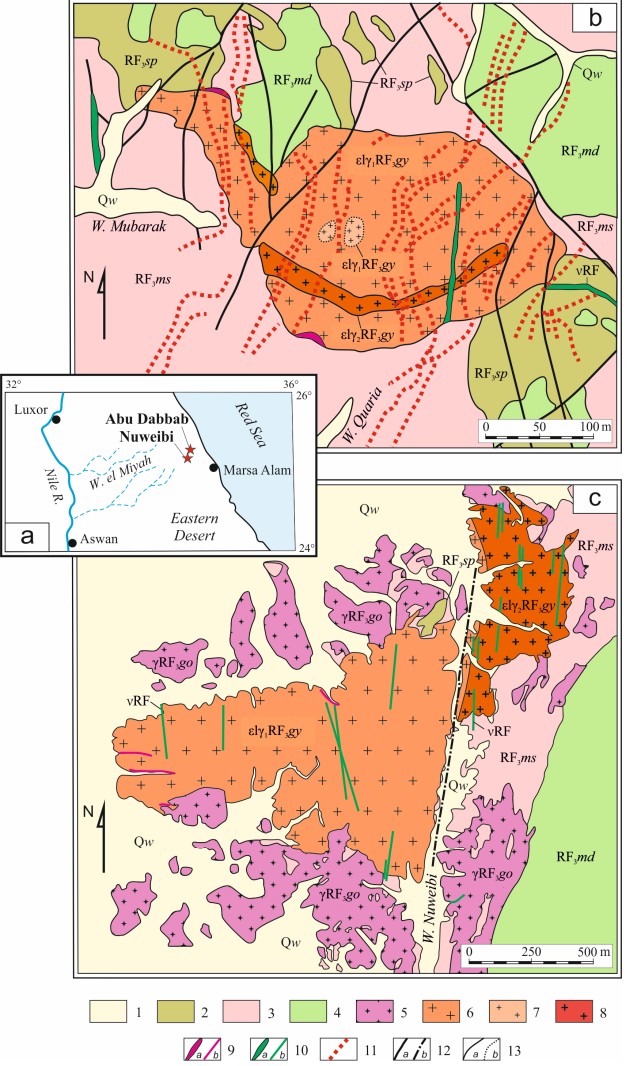

**Figure 1.** Geological map of the Abu Dabbab (**b**) and Nuweibi (**c**) massifs in the Eastern Desert of Egypt (**a**) (adapted from [25] with additions from [12,26,27]). 1—proluvial sand deposits of wadis ($Qw$); 2—serpentinites, talc-carbonate shales ($Rf_3sp$); 3—biotite, hornblende, and chlorite schists ($Rf_3ms$); 4—metagabbro-diorites, metabasalts, meta-andesites ($Rf_3md$); 5—tonalite-granodiorite complex (ancient gray granites) ($\gamma Rf_3go$); 6–9—complex of rare-metal Li-F granites (young albite granites): 6—medium-grained albite granites of the main phase ($\varepsilon l\gamma 1Rf_3gy$), 7—fine-medium-grained albite granites of the main phase ($\varepsilon l\gamma 1Rf_3gy$), 8—porphyritic albite granites of the additional phase ($\varepsilon l\gamma 2Rf_3gy$), 9—stockscheiders with thicknesses of more than 5 m (*a*) and less than 1 m (*b*); 10—dolerite dikes with thicknesses of more than 5 m (*a*) and less than 5 m (*b*); 11—quartz-cassiterite veins; 12—faults: reliable (*a*) and assumed (*b*); 13—geological boundaries between different-age formations (*a*) and facies differences (*b*).

The basement of the region is a Paleoproterozoic granite–gneiss complex, which includes bodies of amphibolites, metagabbro, and a quartz–feldspar schist. In the central part of the Eastern Desert, the basement is overlain by a complex of Neoproterozoic (810–640 million years ago) metapelites and metabasites of the Arabian–Nubian Shield. The Geological Survey of Egypt distinguishes a melange of ophiolites in the region, metagabbro–diorites, metasedimentary and metavolcanic rocks of the green schist facies. The oldest serpentinites and talc-carbonate slates form exotic blocks in the metasedimentary units of sericite, chlorite, and amphibole slates. Above them are metabasalts, metaandesites, and metarhyolites, associated with intrusions of metagabbro, metadiorites, and ancient synorogenic granitoids [1,12]. Terrigenous and volcanogenic Phanerozoic cover deposits are absent in the studied area. The indigenous outcrops of Precambrian rocks are partly covered by quaternary sandy deposits of the desert and wadi valleys [26–28] (Figure 1b,c).

Eastern Egypt is known for the abundance of granitoid intrusions belonging to three plutonic complexes: (1) synorogenic (820–615 million years ago) calc-alkaline I-type granitoids (subduction, pre-collision, "older gray granites"); (2) late Ordovician (610–590 million years ago) calc-alkaline I-type granitoids (collisional, "younger pink granites"); and (3) post-orogenic (620–530 million years ago) subalkaline and alkaline A-type granites (anorogenic, post-collisional, within-plate, "young alkaline granites", rare-metal albite granites). The early complex includes a pluton of granodiorites and tonalites, intersected by the Nuweibi intrusion. The third complex of rare-metal albite granites includes 14 massifs of the Eastern Desert, Umm Naggat, Igla, Nugrus, and others, including the Abu Dabbab and Nuweibi massifs studied in this article [29–31]. The young albite granites are represented mainly by Li-F granites, specialized in Ta, Sn, Nb, Be, and W, as well as alkaline granites (Nb, Zr, Y) and apogranites (Nb, Ta, Sn, Zr, Y, U, Be, W). The rare-metal granite massifs are distinguished by small dimensions (0.1–4 km$^2$) and a characteristic morphology—intrusive domes, lens-shaped deposits, and stocks [32,33]. They are controlled by deep fault zones, represented by independent bodies or are part of large polychronous plutons of calc-alkaline granites [12,34].

### 3.2. Geological Characteristics of the Abu Dabbab Massif

One of the studied objects is the Abu Dabbab massif located 47 km northwest of Marsa Alam in the hilly part of the Eastern Desert. The area of the granite outcrops is 0.06 km$^2$ (Figure 1a). According to the exploration of the eponymous Sn-Ta deposit, the intrusion is close to a harpolith: it dips to the east and to the west-northwest and it transitions to a short wedge-shaped deposit with apophyses and xenoliths of country rocks. The contacts with enclosing biotite and chlorite schists and serpentinites are sharply tempered and accompanied by small (up to several meters) hornfels. The southwest and northwest contacts of the massif are complicated by stockscheiders—druse-like quartz-feldspar albite pegmatoids with fluorite. Two intrusive phases are distinguished in the massif: the main phase of medium-grained albite granites with a facies of fine-medium-grained granites and an additional phase of porphyritic albite granites. The contacts between the phases are not sharp.

In places, the Li-F granites (LFGs) have been altered by post-magmatic amazonitization, albitization, and greisenization. These changes are accompanied by mineralization (Sn, Ta, feldspar) in apogranite areas with sizes of 5–20 m, and metasomatic aureoles near quartz and feldspar veins with thicknesses of 0.05–2 m and lengths of 3–50 m. Groups of sub-parallel veins with north-northeastern extension stretch for 300–500 m, extending beyond the massif into the enclosing strata. Most veins are composed of white coarse-grained quartz, and fluorite–topaz–muscovite selvages contain mineral ores: cassiterite, wolframite, beryl [1,26,35]. The LFGs of the Abu Dabbab massif containing Ta-bearing cassiterite, tantalite-(Mn), columbite-(Mn), microlite, and wodginite are considered as ores [27,34] (Figure 1b).

Based on our petrographic study, the LFGs of the main phase are medium-grained white/light-gray rocks with massive texture that become green-gray amazonite at the periphery. The rock-forming minerals are albite, microcline, quartz, Li-bearing muscovite, and zinnwaldite (Hereinafter "zinnwaldite"—dark trioctahedral mica containing lithium,

similar in composition to siderophyllite-polylithionite series). The sizes of the quartz and alkali feldspar are 1–3 mm. The texture of the LFG is pseudoophitic with an aggregate of differently oriented albite ($An_{0-4}$) with inclusions of round or bipyramidal quartz, tabular microcline, and anhedral mica. Quartz contains oriented poikilitic inclusions of fine albite, muscovite, and topaz, forming the locally observed "snowball" texture. A similar poikilitic structure is observed in microcline–perthite ($Or_{93-99}Ab_{1-7}$). A comparison of quartz inclusion zones from different growth zones using a microprobe showed no difference in the composition of the core albite and the periphery of the "snowball" texture.

The mica is represented by dark-brown Mn-bearing (3.5%–5.5% MnO) zinnwaldite with a size of 0.2–2 mm and a low iron content (8%–10% FeO). White, green-gray, subhedral or anhedral muscovite and Li-bearing muscovite are widely distributed. The accessory minerals are topaz, spessartine, fluorite, cassiterite, tantalite-(Mn), columbite-(Mn), zircon, wodginite, microlite, uranmicrolite, torianite, and uranothorite. Tourmaline, beryl, ilmenite, and allanite have also been described in the LFG [33,34]. The association of accessory minerals with mica is characteristic. Compared to LFG from other areas of the world, the Abu Dabbab massif granites can be distinguished as garnet LFGs.

The porphyritic granites of the additional phase are represented by white porphyritic rocks with fine-grained (0.1–0.5 mm) groundmass. The inclusions are dominated by isometric anhedral quartz with a "snowball" structure and a tabular microcline; in small quantities, anhedral muscovite and skeletal spessartine are present. The main mass is composed of albite, anhedral grains of feldspars, quartz, and mica. The composition of the accessory minerals in the porphyritic LFG does not differ from that of the main phase rocks, but their content is noticeably higher, especially the content of cassiterite and columbite-tantalite. Mn oxide films and dendrites, such as pyrolusite, psilomelane, as well as jacobsite and coronadite, are noted in the cracks in the LFG.

### 3.3. Geological Characteristics of the Nuweibi Massif

The Nuweibi intrusion is located 17.5 km southwest of the Abu Dabbab intrusion and has a similar composition and ore-bearing potential. The intrusion is divided by the Wadi Nuweibi into two parts, West and East Nuweibi, which represent intrusive phases of the main intrusion. West Nuweibi covers an area of 1.0 $km^2$ and is composed of medium-to-coarse-grained microcline-albite granites of the main phase. The contacts with the surrounding ancient granodiorites are almost vertical, with apophyses. East Nuweibi covers an area of 0.26 $km^2$ and consists of white porphyritic albite granite of the additional phase. The contacts with the metamorphic rocks are shallow (15–20°) with apophyses in the surrounding schists, indicating a dome-like shape in the intrusion. The country rocks near the contacts have undergone uneven silicification and amazonitization. The contact between the phases is hidden under the sands of Wadi Nuweibi. In the outcrops of the eastern edge of the valley, a diffuse phase contact of the main phase granite is observed. Geochemical differences have also indicated the affiliation with different phases of the intrusion.

Topaz–zinnwaldite–amazonite–quartz stockscheiders and quartz lenses, with beryl with a thickness of 10–80 cm up to the first meter, are observed at the contacts between microcline–albite granites and schists. At the boundary of the main phase granite and the veins, a zone of striped albite aplite with a thickness of about 10 cm is observed. The main phase granite is disrupted by sub-meridional dolerite dikes. Unlike the Abu Dabbab intrusion, ore-bearing quartz veins are observed only in the host rocks. Veins of a north-northeast orientation have a thickness of 20–30 cm and a length of 100–800 m. The veins and adjacent greisens are composed of Li-micas, topaz, fluorite with inclusions of cassiterite, wolframite, and beryl. The Nuweibi LFGs are Ta ores with accompanying Nb and Sn [12,29,31] (Figure 1c).

Petrographic analysis showed that the main phase granite is a medium-to-coarse-grained (2–5 mm) white, greenish-gray, massive rock with a pseudoophitic texture, and banding in the endocontact zones. The granites are microcline–albite with zinnwaldite and visible inclusions of spessartine and columbite–tantalite. White tabular albite ($An_{1-2}$) with

a length of 0.5–3 mm forms the framework of the rock with inclusions of approximately the same amount of orthoclase and quartz. Orthoclase, with a size of 1–3 mm, is represented by gray microcline–perthite or pale green amazonite ($Or_{96-98}Ab_{2-4}$). Quartz has roundish outlines and a «snowball» structure with zonal inclusions of albite. The micas in the main phase granite are represented by a small amount (~3%) of dark-brown manganese-bearing (0.6%–9.1% MnO) zinnwaldite with low iron content (8%–11% FeO), and greenish-gray muscovite containing Fe and Li. Accessory minerals include columbite-(Mn), tantalite-(Mn), Ta-bearing cassiterite, Hf-bearing zircon, spessartine, topaz, and rare accessory minerals such as wodginite, microlite, thorite, and xenotime. Beryl, apatite, fluorite, monazite, ixiolite, stibiotantalite have been previously described in the main phase granite [29,31].

The porphyritic granite of the additional phase is a white, ovoid, porphyritic rock with a fine-grained (0.05–0.5 mm) groundmass, massive or banded texture. Inclusions, 1–5 mm in size, are represented by pea-shaped or bipyramidal quartz with a "snowball" structure, tabular microcline, rare (1%–2%) anhedral crystals of mica, greenish-gray zinnwaldite, or white lithium muscovite, and single skeletal crystals of spessartine. The matrix of the main phase granite is significantly albitized (~50%) with admixtures of anhedral quartz, microcline, and a small amount of muscovite, enriched in the above-mentioned accessory minerals. The main phase granite of the Nuweibi intrusion contains spotted patches of secondary amorphous Mn oxides and psilomelane along the cracks.

## 4. Results

The minerals associated with LFG from the Abu Dabbab and Nuweibi deposits is formed by topaz, fluorite, spessartine, tantalum–niobates, Ta-bearing cassiterite, fluorapatite, and Hf-bearing zircon.

### 4.1. Morphology and Size of Accessory Wodginite in the Abu Dabbab and Nuweibi Massifs

Wodginite is a common accessory and mineral ore in the studied area. Three forms of the crystallization of wodginite were identified (in order of occurrence): (1) rims in columbite-tantalite; (2) micro-inclusions in cassiterite; (3) autonomous crystals. Rims are usually observed in tantalite-(Mn) crystals or surround tantalite-(Mn) rims in columbite-(Mn) crystals. Contacts between rims, rock-forming silicates, and quartz are sharp, internal contacts with tantalite are fuzzy metasomatic. The thickness of individual rims is constant; the average being 7.8 μm. In different tantalite crystals, wodginite rims vary from 1.5 μm to 21.9 μm (Figure 2a,b), up to the formation of wodginite pseudomorphs with tantalite relict cores (Figure 2b,c). More than 50% of wodginite accessory grains are represented by partial pseudomorphs after tantalite.

Wodginite inclusions in cassiterite are sharply anhedral and angular. They are distributed unevenly in cassiterite and often gravitate towards the periphery of crystals. The size of inclusions is 5.4–27.0 μm; the average being 16.2 μm (Figures 2d,e and 3c). It is likely that autonomous wodginite crystals are partly made up of complete pseudomorphs after tantalite. Euhedral tabular crystals are short and elongated, with a C-axis length of 3–124 μm; the average being 34.6 μm (Figure 2b,f).

In LFG of the Abu Dabbab massif, wodginite enriched in titanium has been found. Its modes of occurrence are similar to those described above, but the rims of Ti-wodginite are dressed with crystals of primary wodginite. The Ti-bearing wodginite rims have indistinct boundaries and varying thickness. In different crystals of wodginite, this thickness ranges from 2.4 to 24.0 μm, with an average of 10.6 μm (Figure 3a,b).

Inclusions of Ti-bearing wodginite in cassiterite are more common than inclusions of low-titanium wodginite, but they are indistinguishable from eachother in morphology and distribution. The size of the inclusions is 8.4–33.8 μm, with a mean of 14.0 μm. Accessory cassiterite with wodginite inclusions is particularly characteristic of porphyritic LFG in the additional phase. Autonomous crystals of Ti-bearing wodginite are idiomorphic tabular, with a length along the C-axis of 3.6–130 μm, with an average of 33.3 μm. Ti-bearing wodginite is clearly associated with Ta-bearing cassiterite (Figure 3c,d). This type of wodginite is often

partially, or almost completely, replaced by microlite. Fe and Ti-bearing wodginite, a rarely occurring mineral phase, has also been found in the main phase of the LFG. It forms anhedral inclusions in cassiterite that range in size from 2.2 to 10.0 μm, with an average of 6.1 μm.

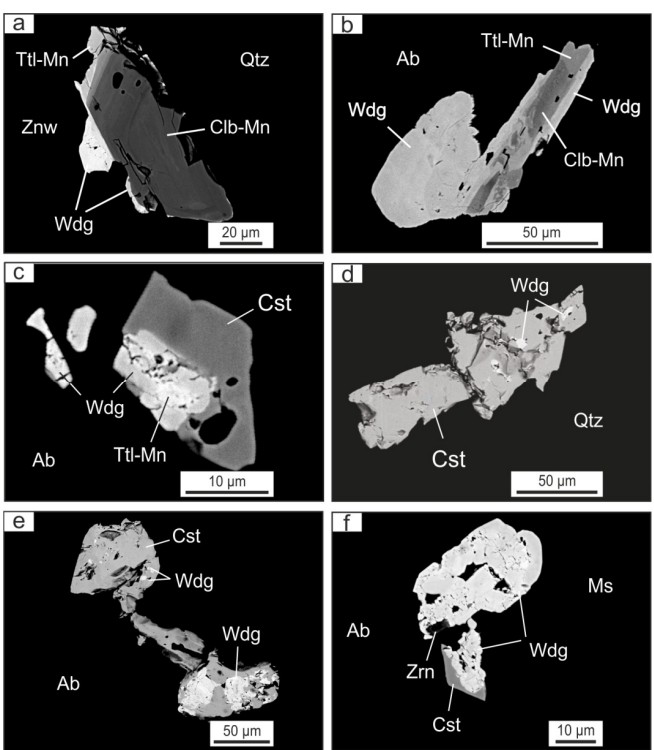

**Figure 2.** Wodginite (Wdg) in lithium-fluorine granites of the Abu Dabbab and Nuweibi massifs, eastern Egypt. Image in backscattered electrons. Mineral indices according to [36]. (**a**) Euhedral oscillatory-zonal columbite-(Mn) with a rim of tantalite-(Mn) and growing wodginite; (**b**) columbite-Mn and tantalite-Mn with a rim and growing wodginite; (**c**) Ta-bearing cassiterite with a pseudomorph inclusion of wodginite after tantalite-(Mn); (**d**,**e**) Ta-bearing cassiterite with micro-inclusions of wodginite; (**f**) tabular wodginite in aggregate with Ta-bearing cassiterite and an inclusion of Hf-U-bearing zircon. (**b**–**d**,**f**)—Abu Dabbab, (**a**,**e**)—Nuweibi.

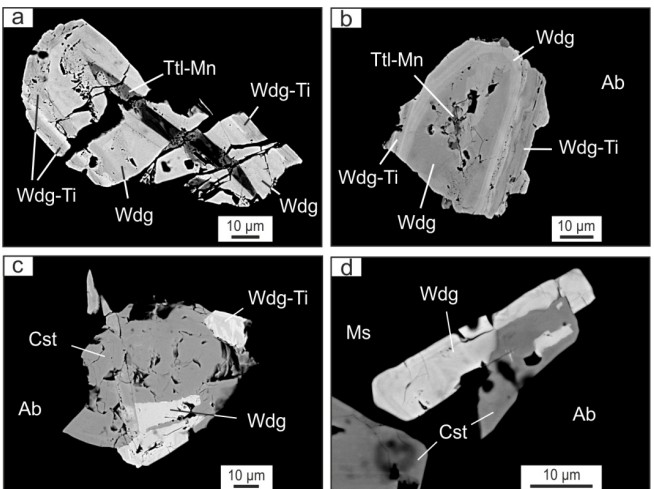

**Figure 3.** Ti-bearing wodginite (Wdg-Ti), wodginite (Wdg) and cassiterite (Cst) in lithium–fluorine granite from the Abu Dabbab massif, Eastern Egypt. Image in backscattered electrons. Mineral indexes according to [36]. (**a**,**b**) encrustation of wodginite with relics of tantalite-(Mn) by Ti-bearing wodginite; (**c**,**d**) growths of Ta-bearing cassiterite with wodginite and Ti-bearing wodginite.

When comparing the wodginite of the main and additional phases, it was established that in the Nuweibi array, wodginite is rarely encountered in LFG in the main phase in the form of tantalite rims. Wodginite of Nuweibi sharply predominates in porphyritic LFG of the additional phase in the form of autonomous crystals and rims in columbite-tantalite. Wodginite and Ti-bearing wodginite from Abu Dabbab are widely distributed as accessory minerals in the LFG in the main and additional intrusive phases.

### 4.2. Composition of Wodginite from the Abu Dabbab and Nuweibi Deposits

Wodginite from the Nuweibi LFG is compositionally similar to wodginite described in other locations [14], but with a lower $TiO_2$ content, averaging at 0.54%; $Ti/(Sn + {}^{B}Ta + Ti + Fe^{3+}) = 0.04$ (Table 1; Figure 4a–c). The empirical formula of Nuweibi wodginite is $(Mn^{2+}_{0.75}Fe^{2+}_{0.25})_{1.00}(Sn_{0.56}Ta_{0.25}Ti_{0.04}Fe^{3+}_{0.19})_{1.04}(Ta_{1.63}Nb_{0.37})_{2.00}O_8$.

**Table 1.** Chemical composition (wt.%) of the wodginite group minerals in Li-F granite of Abu Dabbab and Nuweibi massifs.

| Component | Abu Dabbab | | | | | | | | | Nuweibi | | |
| | Wodginite | | | Ti-Bearing Wodginite | | | Fe-Ti-Bearing Wodginite | | | Wodginite | | |
| | A-31 | A-72 | Av. | A-27 | A-64 | Av. | A-28 | A-62 | Av. | N-4 | N-17 | Av. |
|---|---|---|---|---|---|---|---|---|---|---|---|---|
| MnO | 8.01 | 8.18 | 9.32 | 10.53 | 12.12 | 10.77 | 6.49 | 6.11 | 6.30 | 7.90 | 8.67 | 8.39 |
| FeO$_t$ [1] | 3.79 | 4.50 | 3.40 | 1.34 | 1.04 | 1.58 | 5.92 | 6.48 | 6.93 | 5.78 | 4.77 | 4.91 |
| TiO$_2$ | 0.00 | 0.37 | 1.00 | 6.00 | 2.94 | 3.03 | 2.41 | 4.55 | 3.48 | 0.42 | 0.47 | 0.54 |
| Nb$_2$O$_5$ | 6.26 | 7.60 | 8.26 | 8.99 | 13.07 | 8.88 | 12.44 | 13.55 | 13.00 | 5.80 | 7.96 | 7.63 |
| SnO$_2$ | 15.38 | 14.54 | 14.50 | 11.62 | 12.83 | 13.26 | 13.80 | 9.66 | 11.73 | 9.64 | 13.43 | 13.20 |
| Ta$_2$O$_5$ | 65.20 | 64.53 | 63.41 | 61.04 | 57.99 | 61.07 | 59.09 | 58.40 | 58.75 | 70.13 | 64.56 | 65.33 |
| Total | 98.64 | 99.71 | 99.89 | 99.52 | 99.99 | 98.57 [2] | 100.20 | 98.75 | 99.48 | 99.67 | 99.85 | 100.01 |
| Structural formula (apfu) calculated on the basis of O = 8 atoms [3] | | | | | | | | | | | | |
| Mn | 0.74 | 0.74 | 0.83 | 0.89 | 1.03 | 0.95 | 0.55 | 0.52 | 0.54 | 0.72 | 0.78 | 0.75 |
| Fe$^{2+}$ | 0.26 | 0.26 | 0.17 | 0.11 | 0.00 | 0.05 | 0.45 | 0.48 | 0.46 | 0.28 | 0.22 | 0.25 |
| Σ A | 1.00 | 1.00 | 1.00 | 1.00 | 1.03 | 1.00 | 1.00 | 1.00 | 1.00 | 1.00 | 1.00 | 1.00 |
| Fe$^{3+}$ | 0.08 | 0.14 | 0.13 | 0.00 | 0.11 | 0.08 | 0.05 | 0.06 | 0.06 | 0.25 | 0.20 | 0.19 |
| Ti | 0.00 | 0.03 | 0.08 | 0.45 | 0.22 | 0.24 | 0.18 | 0.34 | 0.26 | 0.03 | 0.04 | 0.04 |
| Sn | 0.67 | 0.62 | 0.61 | 0.46 | 0.51 | 0.55 | 0.55 | 0.38 | 0.47 | 0.41 | 0.57 | 0.56 |
| Ta | 0.23 | 0.23 | 0.20 | 0.07 | 0.17 | 0.14 | 0.19 | 0.20 | 0.19 | 0.34 | 0.24 | 0.25 |
| Σ B | 0.98 | 1.02 | 1.02 | 0.98 | 1.01 | 1.01 | 0.97 | 0.98 | 0.98 | 1.03 | 1.05 | 1.04 |
| Nb | 0.31 | 0.36 | 0.39 | 0.41 | 0.59 | 0.42 | 0.57 | 0.61 | 0.59 | 0.28 | 0.38 | 0.37 |
| Ta | 1.70 | 1.63 | 1.61 | 1.59 | 1.41 | 1.58 | 1.43 | 1.38 | 1.41 | 1.72 | 1.62 | 1.63 |
| Σ C | 2.01 | 1.99 | 2.00 | 2.00 | 2.00 | 2.00 | 2.00 | 1.99 | 2.00 | 2.00 | 2.00 | 2.00 |

The representative electron probe microanalysis and average composition (Av.) of wodginite, Ti-bearing wodginite, Fe and Ti-bearing wodginite from Abu Dabbab (25, 75, 2 samples, respectively), and of wodginite from Nuweibi (27 samples) is presented. [1] FeOt = FeO + Fe$_2$O$_3$. [2] Including CaO 0.01. [3] Values are calculated using stoichiometry; the cations of Fe$^{2+}$, and Fe$^{3+}$ calculated according to the methods of [17].

We have divided the Abu Dabbab wodginite into three types: "normal" wodginite, Ti-bearing wodginite, and Fe and Ti-bearing wodginite. Compared to the global wodginite, wodginite from Abu Dabbab is enriched in manganese and tin, with an average $Mn/(Mn + Fe^{2+}) = 0.83$ and $Sn/(Sn + {}^{B}Ta + Ti + Fe^{3+}) = 0.60$ (Table 1; Figure 4a,b). No significant differences were observed between the wodginite in the main and accessory phases of LFG in Abu Dabbab. The empirical formula of Abu Dabbab wodginite is $(Mn^{2+}_{0.83}Fe^{2+}_{0.17})_{1.00}(Sn_{0.61}Ta_{0.20}Ti_{0.08}Fe^{3+}_{0.13})_{1.02}(Ta_{1.61}Nb_{0.39})_{2.00}O_8$.

Ti-bearing wodginite is classified as being similar to titanowodginite, with $TiO_2$ content $\geq$ 2.00% (~0.15 Ti atoms per formula unit); on average, $Ti/(Sn + {}^{B}Ta + Ti + Fe^{3+}) = 0.23$ (Table 1). The boundary between wodginite and Ti-bearing wodginite on the Mn/ΣA–Sn/ΣB diagram is set at an Sn content of 0.3 atoms per formula unit, with, on average, 0.2 formula units occupied by Ta atoms (Figure 4b). This variety of wodginite is maximally enriched in manganese: $Mn/(Mn + Fe^{2+} + Ca) = 0.95$, and depleted in tantalum: $Ta/(Ta + Nb) = 0.81$ (Figure 4a–c). The highest proportion of tantalum, typical of global wodginite, is observed in the Ti-bearing wodginite of porphyritic LFG in the additional phase: $Ta/(Ta + Nb) > 0.80$. According to the

Sn and Ti content, this variety of wodginite in the main and additional phases of Abu Dabbab LFG does not differ. The empirical formula is $(Mn^{2+}_{0.95}Fe^{2+}_{0.05})_{1.00}(Sn_{0.55}Ta_{0.14}Ti_{0.24}Fe^{3+}_{0.08})_{1.01}$ $(Ta_{1.58}Nb_{0.42})_{2.00}O_8$.

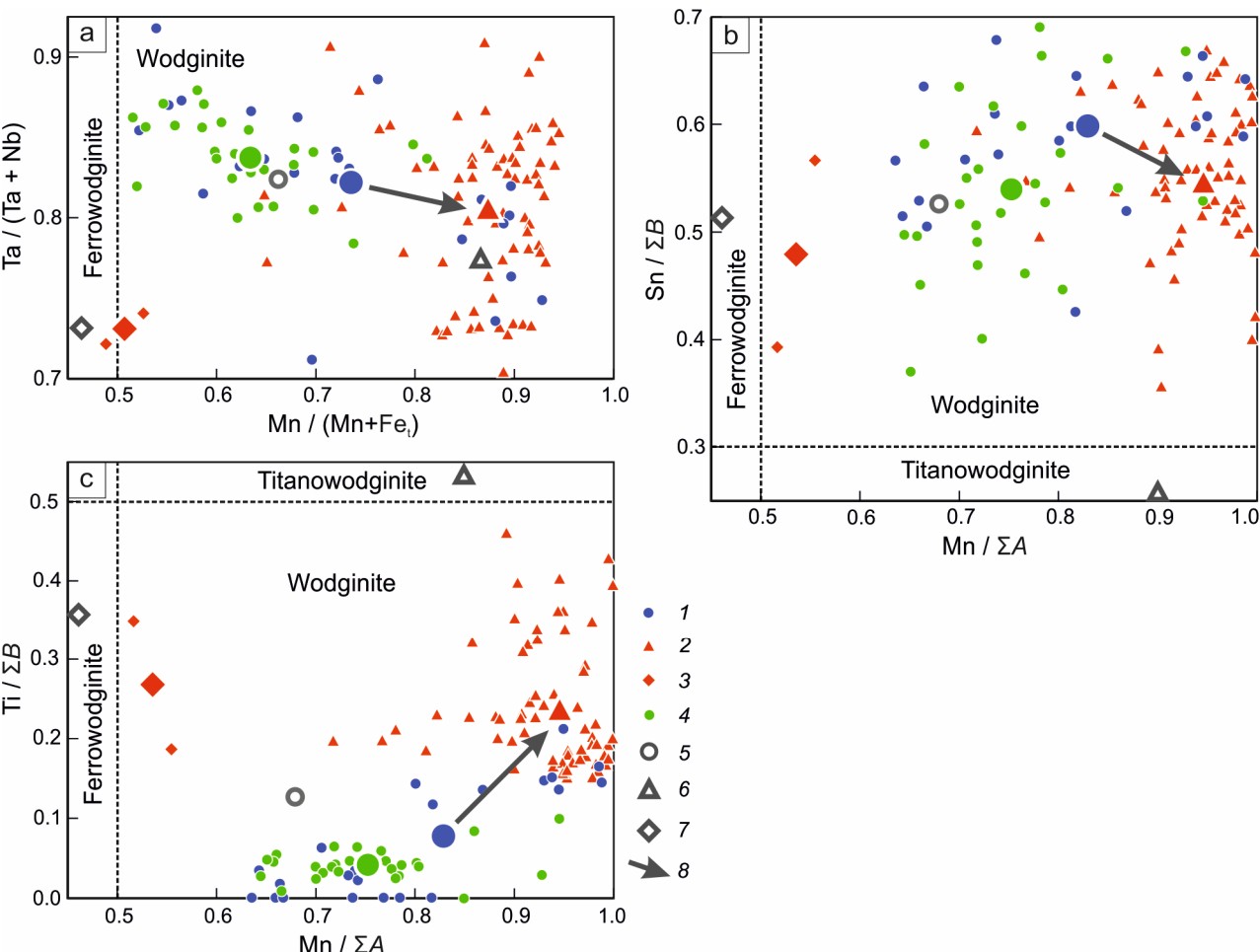

**Figure 4.** Ratios of the principal cations in minerals of the wodginite group from lithium–fluorine granites from the Abu Dabbab and Nuweibi massifs. Diagrams of wodginite classification: (**a**) Mn/(Mn+Fe) versus Ta/(Ta+Nb), (**b**) Mn/$\Sigma A$ versus Sn/$\Sigma B$, (**c**) Mn/$\Sigma A$ versus Ti/$\Sigma B$. The atomic ratios of the cations in wodginite: (1), Ti-bearing wodginite (2), and Fe-Ti-bearing wodginite (3) of Abu Dabbab granites, and wodginite of Nuweibi granites (4) are presented. The large symbols of the same color indicate the average compositions of the corresponding minerals. Unfilled symbols represent the world-wide averages of wodginite-group minerals according to [14]: (5)—wodginite, (6)—titanowodginite, (7)—ferrotitanowodginite. The arrow (8) illustrates the evolution of wodginite in the Abu Dabbab granites. $Fe_t = Fe^{2+} + Fe^{3+}$. $\Sigma A$, $\Sigma B$—sums of cations in *A*-position $(Mn^{2+}, Fe^{2+}, Ca)$, *B* $(Sn, {}^B Ta, Ti, Fe^{3+})$.

A small proportion (2%) of wodginite in the Abu Dabbab massif contains an increased ratio of Fe and Ti cations and is virtually equivalent to ferrotitanowodginite (Figure 4a–c). Fe and Ti-bearing wodginite is defined by the saturation of $Fe^{2+}$ cations in *A*-position: 0.46 atoms per formula unit, FeO 5.92%–6.48%; and the increased content of $TiO_2$ 2.41%–4.55%; 0.26 atoms per formula unit (Table 1). Tantalum–niobium associates with wodginite contain tin impurity: columbite-(Mn) $SnO_2$ up to 1.83% (on average 0.15%, 24% of samples); tantalite-(Mn) $SnO_2$ up to 2.51% (on average 0.45%, 48% of samples); microlite $SnO_2$ up to 4.86% (on average 0.61%, 18% of samples).

## 5. Discussion

### 5.1. Genesis of Wodginite

Wodginite in the LFGs of Abu Dabbab and Nuweibi is associated with the main accessory minerals of tantalum—tantalite-(Mn), cassiterite, and microlite. The formation of tantalum-niobates in several stages, the complex zoning, and the association with cassiterite are characteristic features. Wodginite forms in the late stages through the overgrowth and replacement of earlier tantalite-(Mn) (Figures 2 and 3). This corresponds to the existing concepts of wodginite as a marker of the transition from late-magmatic crystallization of rare-metal granitic magma to pneumatolitic mineralization [10,13,37]. The normal course of crystallization of tantalum niobates is determined by the increase in Ta/(Ta + Nb) and Mn/(Mn + Fe): columbite-(Fe) → columbite-(Mn) → tantalite-(Mn) → wodginite → microlite [38–40]; in tin-bearing fluid-magmatic systems: columbite-(Mn) → tantalite-(Mn) → wodginite + titanowodginite → cassiterite [13].

The combination of different mineral species of the same elements, the abundance of mineral inhomogeneities, and the variety of impurity elements indicate a long history of mineralization and the geochemical richness of ore sources, which is typical of large deposits of mineral raw materials. The study of wodginite in the LFGs of eastern Egypt confirms the previous conclusion that wodginite is an indicator of ore-bearing granites and associated tantalum and tin mineralization [14]. For example, the presence of wodginite with similar characteristics in alluvial deposits is an indicator of the native tin and tantalum-bearing LFG and greisens.

Three forms of wodginite crystallization are identified in the LFG, from which wodginite rims in the accessory tantalite and autonomous wodginite crystals (Figures 2a–c,f and 3a,b,d) indicate the metasomatic replacement of tantalite-(Mn) by wodginite in the presence of a late-magmatic fluid phase [14,41]. The inclusion of wodginite in pegmatite cassiterite led researchers to conclude that they formed during the breakdown of the solid solution "cassiterite–tantalum-niobates" [42]. Inclusions of wodginite and Ti-bearing wodginite in Egyptian LFG (Figures 2d,e and 3c) are similar to the described inclusions of wodginite in depleted cassiterite [43].

### 5.2. Isomorphism and Genetic Significance of Titanium-Bearing Wodginite

Wodginite is characterized by a perfectly layered structure, $ABC_2O_8$, determined by the presence of Sn cations (position $B$) and the excess of Ta (position $C$) and Mn (position $A$) cations in the rare-metal-bearing melt that forms the mineral. The main isomorphism scheme is $^A[Fe^{2+},Mn]^{2+} + 2^C[Nb,Ta]^{5+} \leftrightarrow 3^B[Sn,Ti]^{4+}$ [14,38]. As seen from Table 1, there is a charge imbalance in position $B$ caused by the introduction of Ta, which generates vacancies in position $A$: $^A\square + 2^B[Ta^{5+}] \leftrightarrow {}^A[Mn^{2+}] + 2^B[Sn^{4+}]$; as well as isomorphism: $2^B[Sn^{4+}] \leftrightarrow {}^B[Fe^{3+}] + {}^B[Ta^{5+}]$ [17,40]. The most effective and significant isomorphism in position $B$ is Ti ↔ Sn; 52%–75% of the Sn can be replaced by Ti [17]. Previous mineral association of wodginite and titanowodginite have been described in rare-metal granites of Primorye (Russia) [5], China [3,9,11], and Algeria [13]. Ferrotitanowodginite has been described in the LFGs of the Gedongping massif (China) [11]. There has been a suggestion of mixing discontinuity between wodginite $(Fe,Mn)SnTa_2O_8$ and titanowodginite $(Fe,Mn)TiTa_2O_8$ [9]. According to [39], the origin of titanowodginite in rare-metal rocks occurs during the transition from magmatic fractionation to post-magmatic fluid activity.

Ti-bearing wodginite overwhelmingly predominates in Abu Dabbab granite (Table 1), replacing wodginite (Figure 4a–c). It is probably that a late-magmatic enrichment of fluids with tin and titanium in equilibrium with residual melt and abundant crystallization of cassiterite determined the replacement of tin by titanium in the late-magmatic wodginite. Thus, the well-known sequence columbite → tantalite → wodginite is supplemented in tin-bearing deposits by the stage wodginite → Ti-bearing wodginite + Fe and Ti-bearing wodginite. The significant role of $Fe^{3+}$ cations indicates oxidizing crystallization conditions in LFGs, promoting the crystallization of cassiterite. It has been established that, during the breakdown of the solid solution of cassiterite containing titanium impurity, not only

wodginite inclusions but also Ti-bearing wodginite inclusions can occur. Although tin is in excess, Ti is involved in the formation of the wodginite lattice, isomorphously replacing Sn in position *B*.

A comparison of wodginite in the two studied LFG massifs shows that wodginite in Nuweibi: (1) occurs mainly in the porphyry granites of the additional phase; (2) is close to the worldwide wodginite composition of rare-metal granites; (3) has a reduced $TiO_2$ content; (4) is a mineral indicator of rich tantalum mineralization [12]. Unlike Nuweibi, wodginite in Abu Dabbab is widely represented in the main and additional phases of the granitic massif, predominantly in the titanian variety (Table 1, Figure 3c). At the same time, the Nuweibi massif does not contain greisen tin mineralization, while the Abu Dabbab massif is rich in greisens and quartz veins with cassiterite [1,29,31] (compare Figure 1b,c). It is essential to note that the increase in $TiO_2$ impurity is characteristic of greisen cassiterite in comparison with accessory cassiterite of tantalum-bearing granites [44].

It can be concluded that the indicators of tantalum and associated tin mineralization are: (1) the widespread occurrence of wodginite in all intrusive phases of the LFG; (2) the presence of accessory cassiterite with inclusions of wodginite in the LFG; (3) the very likely, the presence of Ti-bearing wodginite as a separate phase, accompanying wodginite. The indicator of a rich tantalum mineralization is autonomous accessory wodginite and wodginite rims in the tantalite of the LFG additional phase.

## 6. Conclusions

1. Accessory minerals of the wodginite group, wodginite, Ti-bearing wodginite, and rare Fe and Ti-bearing wodginite associated with Ta-bearing cassiterite and Sn-bearing tantalum-niobates, are distributed in the lithium–fluorine granites of the Eastern Desert of Egypt.

2. Three forms of wodginite crystallization are identified: (1) rims in columbite–tantalite; (2) micro-inclusions in cassiterite; (3) autonomous crystals. The average size of wodginite crystals is 34.6 μm. Wodginite crystals represent partial or complete pseudomorphs after tantalite-(Mn). Wodginite is marker of the transition from late-magmatic crystallization of rare-metal granitic magma to pneumatolitic mineral formation in the presence of a fluid phase.

3. Wodginite of the Nuweibi massif mainly occurs in the porphyritic granites of an additional granitic phase. It is close in composition to the worldwide wodginite of rare-metal granites, is distinguished by a reduced $TiO_2$ content, and is a mineral indicator of rich tantalum mineralization.

4. Wodginite in the Abu Dabbab massif is replaced by Ti-bearing wodginite, and both types are maximally enriched with Mn. They are represented in all intrusive phases of the massif, and are very likely to be mineral indicators of tantalum-bearing granites with accompanying cassiterite–quartz mineralization.

**Author Contributions:** Conceptualization, V.I.A.; methodology, V.I.A.; software, V.I.A.; validation, V.I.A.; formal analysis, V.I.A.; investigation, V.I.A.; resources, V.I.A.; data curation, V.I.A. and I.V.A.; writing—original draft preparation, V.I.A.; writing—review and editing, I.V.A.; visualization, V.I.A. and I.V.A. All authors have read and agreed to the published version of the manuscript.

**Funding:** This research received no external funding.

**Data Availability Statement:** The data presented in this study are available on request from the corresponding author. The data are not publicly available due to copyright of St. Petersburg Mining University.

**Acknowledgments:** The authors extend appreciation to the employees of the Institute of Precambrian Geology and Geochronology, Russian Academy of Sciences, and to the employees of Saint-Petersburg Mining University for analytical studies.

**Conflicts of Interest:** The authors declare no conflict of interest.

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
