# Peer review of "The Presence of Wodginite in Lithium–Fluorine Granites as an Indicator of Tantalum and Tin Mineralization: A Study of Abu Dabbab and Nuweibi Massifs (Egypt)"

_minerals, doi:10.3390/min13111447_

Round 1

Reviewer 1 Report

Comments and Suggestions for Authors

This paper presents the results of investigation the geochemistry of accessory wodginite from  tantalum-bearing granites of Abu Dabbab and Nuweibi massifs in Eastern Egypt as indicator of  tantalum and tin mineralization in granites. The authors rightly point out that the results of their research can help in solving a number of questions of the Ta and Sn mineralization. The main attention in the article was directed to the study of the mechanism of Fe, Ti Nb, and Ta incorporation into the wodginite structure. The main conclusions of the authors are based on their own analytical data obtained using modern equipment (electron probe micro-analysis, XRF-1800 X-ray fluorescence spectrometry and ICPE-9000 inductively coupled plasma mass spectrometry) and processed by methods of mathematical statistics. The data interpretations seem to have been done well, also the data obtained may be valuable for understanding the geochemical features of wodginite as the indicator of a кich tantalum deposits. So, the paper adds valuable new information concerning our knowledge on the incorporation mechanisms of Ta and accompanying elements into the wodginite under consideration. I thus can recommend the paper for publication in Minerals but with minor revisions.

Comments and noticed typos

Line 9 and etc. “ferrotitanowodginite”  = Fe and Ti-bearing wodginite (because it is variety of wodginite)

Line 41. Tanalowodginite = Tantalowodginite

Line 155. Ta-cassiterite = Ta-bearing cassiterite

Line 293. Ti-wodginite = Ti-bearing wodginite

Line 307. …of the wodginite group minerals = wodginite (because in the table shows only analyses of wodginite and its varieties.)

Line 323-324. The commas in the formula must be removed: (Mn2+0.95Fe2+0.05)1.00(Sn0.55Ta0.14Ti0.24Fe3+0.08)1.01(Ta1.58Nb0.42)2.00O8.

Reviewer 2 Report

Comments and Suggestions for Authors

-          Who named Abu Dabbab granites as by Li-F granites? Please, add references.

-          Please, add field and some microscopic photos showing snowball and other textures.

-          Please, cite the following reference which studied the Abu Dabab granites: Abuamarah, B.A., Azer, M.K., Asimow, P.D., Ghrefat, H. and Mubarak, H.S., 2021. Geochemistry and petrogenesis of late Ediacaran raremetal albite granites of the ArabianNubian Shield. Acta Geologica SinicaEnglish Edition, 95(2), pp.459-480.

Reviewer 3 Report

Comments and Suggestions for Authors

This study provides a good example of wodginite from lithium-fluorine granites (LPG) Abu Dabbab and Nuweibi occurring in multiple textural contexts and with different compositional groups. 

1) The evidence presented strongly supports the different wodginite compositional domains for each intrusion. The authors try to link the distribution of wodginite and its Ti-bearing composition in the Abu Dabbab intrusion with the surrounding greisen system and ensuing Sn mineralization, whereas the Nuweibi intrusion with its different wodginite signature has no Sn mineralization.
However, the direct relation of the wodginite composition and distribution with the mineralization potential is not so certain. It can indeed be a correlation, but it could also be a coincidence. Other Sn-mineralized regions commonly show Ti-poor wodginite (e.g., Melcher et al. 2015. OGR, 64, 667-719), although, to my knowledge, not many studies have given so much detail to wodginite.
It would interesting to assess if a third intrusion (from the many LPGs around the study area) confirms the author’s interpretation. Does another Ta-Sn mineralized granite-greisen system also show Ti-bearing wodginite?

2) I strongly disagree with the nomenclature “ferrotitanowodginite”. To be classified as ferrotitanowodginite (Galliski et al. 1999. Am. Min, 84(5-6), 773-777), the mineral should have a minimum of 50% filling of Ti in the B crystallographic site (0.5 apfu of Ti on the basis of 8 oxygens) and 50% of Fe2+ in the A site (0.5 apfu of Fe2+). These requisites are not fulfilled in the presented chemical analyses. The authors should stick with Fe-Ti-bearing (or Fe-Ti-rich) wodginite, as it was used in the legend of Figure 4.

3) There is some confusion with the term main and additional intrusive phase (in relation to the magmatic stages of each massif). The terms secondary, accessory, and additional are used interchangeably and it is very hard for the reader to understand what the text is referring to. I suggest choosing only one term and applying the same term all over the manuscript.

4) Lastly, the paragenesis of the described phases is not very clear. One genetic relation is hinted at in the results section (tantalite –> wodginite), whereas in the discussion, a different and unsupported relation (tantalite and wodginite in stable paragenesis) is put forward. The discussion section 4.1 (genesis of wodginite), as it is now, does not correlate with the presented results. I suggest the authors reassess their paragenetic relations and rewrite the discussion to be in agreement with the results. Furthermore, the study would benefit significantly from a more in-depth investigation of the paragenesis of each wodginite textural domain (also including other non-oxide mineral phases).

Comments on the Quality of English Language

Some sentences and paragraphs (e.g., lines 53-66) are very confusing. I suggest the authors break the long sentences into smaller segments to facilitate the reading. Additionally, some minor typos and missing words occur in the text.

Please see the marked manuscript for more specific suggestions.

Round 2

Reviewer 3 Report

Comments and Suggestions for Authors

I thank the authors for considering my comments and correcting most of them.

I accept the replies from the authors and think that the manuscript is much clearer now.

However, I'm not convinced by the reply to my comment 4 (genesis of wodginite). I appreciate that the authors changed the term paragenesis to mineral association, as this is more correct in this context. Nevertheless, the main issue is not solved.

The authors claim in section 5.1 that wodginite and tantalite form a STABLE mineral association (373-375). However, in the results section, they clearly claim that wodginite occurs REPLACING tantalite as pseudomorphs (lines 274-276 and Fig. 2).
Replacement textures are a strong sign of disequilibrium. Two phases with replacement textures between them, cannot represent a stable mineral association.
The authors went even further this round and removed the term replacement from the discussion (line 378), giving the impression that wodginite precipitates directly from the melt, only using tantalite as a growing surface (overgrowth) in a continuous process.
My initial comment was intended to highlight the requirement of a fluid phase in the formation of wodginite (as is also interpreted for Sn mineralization). As pointed out by the authors, Wdg is a good marker of the magmatic-hydrothermal transition (lines 379-381). My apologies if my initial comment was not clear enough.

In summary, I think that the authors should be consistent in their model for the genesis of Wdg, and their reply and modifications did not solve this issue. Wdg is either stable with tantalite or formed by the replacement of the latter in the presence of a fluid phase.
My interpretation, looking at the presented data, is that the second option is more correct. Therefore, the authors should point out more clearly the role of the fluid phase (and the magmatic-hydrothermal transition) in the genesis of wodginite.
